# The Effects of Long-Term High Fat and/or High Sugar Feeding on Sources of Postprandial Hepatic Glycogen and Triglyceride Synthesis in Mice

**DOI:** 10.3390/nu16142186

**Published:** 2024-07-09

**Authors:** Ana Reis-Costa, Getachew D. Belew, Ivan Viegas, Ludgero C. Tavares, Maria João Meneses, Bárbara Patrício, Amalia Gastaldelli, Maria Paula Macedo, John G. Jones

**Affiliations:** 1PhD Programme in Experimental Biology and Biomedicine, Institute for Interdisciplinary Research, University of Coimbra, 3030-789 Coimbra, Portugal; anamrcosta96@gmail.com (A.R.-C.); getachew.debas@ohio.edu (G.D.B.); 2Center for Neuroscience and Cell Biology (CNC-UC), Institute for Interdisciplinary Research, University of Coimbra, 3030-789 Coimbra, Portugal; 3Grupo de Estudos de Investigação Fundamental e Translacional (GIFT) da Sociedade Portuguesa de Diabetologia, 1250-198 Lisboa, Portugal; 4Department of Biomedical Sciences, Heritage College of Osteopathic Medicine, Ohio University, Athens, OH 45701, USA; 5Centre for Functional Ecology (CFE), TERRA Associate Laboratory, Department of Life Sciences, University of Coimbra, 3030-790 Coimbra, Portugal; iviegas@uc.pt; 6Vasco da Gama Research Center (CIVG), University School Vasco da Gama, 3020-210 Coimbra, Portugal; ludgero.tavares@euvg.pt; 7iNOVA4Health, NOVA Medical School, Faculdade de Ciências Médicas, Universidade NOVA de Lisboa, 1150-082 Lisboa, Portugal; mariajoaocmo@gmail.com (M.J.M.); barbaragomespatricio@gmail.com (B.P.); paula.macedo@nms.unl.pt (M.P.M.); 8National Research Council (CNR), Institute of Clinical Physiology (IFC), 56124 Pisa, Italy; amalia.gastaldelli@cnr.it; 9Scuola Superiore Sant’Anna, 56127 Pisa, Italy; 10APDP-Diabetes Portugal Education and Research Center (APDP-ERC), 1250-203 Lisboa, Portugal

**Keywords:** fructose, acetyl-CoA, Indirect pathway, ^13^C-isotopomer, lipogenesis

## Abstract

Background: In MASLD (formerly called NAFLD) mouse models, oversupply of dietary fat and sugar is more lipogenic than either nutrient alone. Fatty acids suppress de novo lipogenesis (DNL) from sugars, while DNL inhibits fatty acid oxidation. How such factors interact to impact hepatic triglyceride levels are incompletely understood. Methods: Using deuterated water, we measured DNL in mice fed 18-weeks with standard chow (SC), SC supplemented with 55/45-fructose/glucose in the drinking water at 30% (*w*/*v*) (HS), high-fat chow (HF), and HF with HS supplementation (HFHS). Liver glycogen levels and its sources were also measured. For HS and HFHS mice, pentose phosphate (PP) fluxes and fructose contributions to DNL and glycogen were measured using [U-^13^C]fructose. Results: The lipogenic diets caused significantly higher liver triglyceride levels compared to SC. DNL rates were suppressed in HF compared to SC and were partially restored in HFHS but supplied a minority of the additional triglyceride in HFHS compared to HF. Fructose contributed a significantly greater fraction of newly synthesized saturated fatty acids compared to oleic acid in both HS and HFHS. Glycogen levels were not different between diets, but significant differences in Direct and Indirect pathway contributions to glycogen synthesis were found. PP fluxes were similar in HS and HFHS mice and were insufficient to account for DNL reducing equivalents. Conclusions: Despite amplifying the lipogenic effects of fat, the fact that sugar-activated DNL *per se* barely contributes suggests that its role is likely more relevant in the inhibition of fatty acid oxidation. Fructose promotes lipogenesis of saturated over unsaturated fatty acids and contributes to maintenance of glycogen levels. PP fluxes associated with sugar conversion to fat account for a minor fraction of DNL reducing equivalents.

## 1. Introduction

Metabolic Dysfunction-Associated Steatotic Liver Disease (MASLD), formerly known as Non-Alcoholic Fatty Liver Disease (NAFLD), is a highly prevalent co-morbidity of obesity and has become the most prevalent liver disease worldwide [1,2,3]. The most recent epidemiologic study indicates that 30.1% of the world’s population had MASLD between 1990 and 2019. Moreover, an increase of 50.4% was observed between (1990–2006) and (2016–2019), and Latin America has become the most affected region of the globe with 44.4% prevalence [4].

While intrinsic unmodifiable factors such age, sex, ethnicity, geography and genetic polymorphisms all contribute to MASLD risk [4,5,6,7], it is considered that excessive intake of food/s and drinks rich in fat and sugar coupled with a sedentary lifestyle is the principal driver of surging MASLD rates [8,9,10]. Although the liver is a key nexus for both lipid and carbohydrate metabolism, under normal circumstances its lipid content is negligible compared to that of the whole body. For example, a 1.5 kg liver of a healthy adult with 1–2% lipid content represents just 0.015–0.030 kg of triglyceride.

As shown in Figure 1, hepatic triglyceride accumulation is caused by an excess of hepatic lipid inflow and/or de novo synthesis relative to lipid outflow as very-low density lipoprotein (VLDL), and/or fatty acid oxidation. This can be promoted either by restricting triglyceride outflow, for example by providing a diet deficient in choline and methionine that attenuates synthesis of the VLDL protein component and/or by increasing lipid inflow and/or de novo synthesis through high-fat or high-sugar feeding [3,11,12,13]. The Westernized diets that are implicated in MASLD have excesses of both fat and refined sugars and this formulation is being increasingly applied to rodent diets, because the high fat/sugar combination provokes more severe MASLD than either high-fat or high-sugar feeding alone [14,15]. At the level of hepatic metabolic fluxes, high-fat and high-sugar diets induce MASLD in distinctive ways. With diets that have a high proportion of lipid, increased inflows of non-esterified fatty acids (NEFA) from both diet and peripheral adipose tissue lipolysis are considered to be the main drivers of triglyceride accumulation with synthesis of fatty acids via DNL being a relatively minor contributor—partly due to fatty acid suppression of glucose oxidation to acetyl-CoA [16,17,18]. In contrast, a diet that has high sugar relative to lipid levels results in increased DNL activity that not only directly contributes with fatty acid moieties to the hepatic lipid load but also suppresses the β-oxidation of long-chain fatty acids via its malonyl-CoA intermediate [19,20]. For diets that are abundant in both sugar and lipid, it is unclear to what extent the reciprocal controls between oxidation and biosynthesis of carbohydrates and lipids are maintained. These may be further undermined when the dietary sugar has a high proportion of fructose—a monosaccharide whose conversion to lipid bypasses key control points of carbohydrate catabolism [21]. In many Western countries, the replacement of sucrose by high-fructose corn syrup (HFCS) and the prevalence of so-called “low fat” processed foods that are rich in sugar has further increased fructose intake [22,23,24,25].

Fructose metabolism is independent of insulin signaling, takes place mostly in the liver and bypasses the rate limiting step of glycolysis catalyzed by phosphofructokinase-1. In this way, fructose is rapidly phosphorylated to fructose 1-phosphate via fructokinase without facing any feedback control loop, thus becoming more lipogenic than glucose [26,27,28]. Fructose- and sucrose-sweetened beverages promote de novo fatty acid synthesis, especially palmitate, without upregulating VLDL secretion, therefore potentiating ectopic fat deposition [29]. Nonetheless, according to Smajis et al., young, healthy subjects show no significant increase in liver fat or postprandial glycogen stores following 8 weeks of high-fructose supplementation partly due to temporary compensation of the calories obtained from other macronutrients [30]. Pediatric MASLD patients absorb and metabolize more fructose compared to both lean- and obese- matched subjects. Moreover, fructose intake led to greater plasma glucose, insulin and uric acid concentrations in the same patients [31].

Hepatic glycogen is a critical short-term store of endogenous carbohydrate and may also play a role in hepatic fuel sensing [32,33]. Many of the endogenous factors that regulate lipid biosynthesis from carbohydrate—such as glucokinase activation and insulin signaling—also influence the recruitment of glucose and gluconeogenic substrates for glycogen synthesis. Moreover, dietary fructose independently influences hepatic glycogen synthesis both as a gluconeogenic substrate (Indirect pathway) and as a glucokinase activator [21,34]. To date, the effects of excessive dietary sugar, fat, or both, on the sources of hepatic glycogen synthesis have not been characterized.

In this study, we applied recently developed methods for quantifying substrate fluxes into hepatic lipid and glycogen in mice during overnight ad libitum feeding to evaluate the influence of an extended interval of high-fat, high-sugar and high-fat plus high-sugar feeding on the sources of hepatic lipid and glycogen synthesis [35,36,37].

## 2. Methods

### 2.1. Animal Studies

Animal studies were approved by the University of Coimbra Ethics Committee on Animal Studies (ORBEA) and the Portuguese National Authority for Animal Health (DGAV, 0421/000/000/2017). All animal procedures were performed according to DGAV guidelines as well as European regulations (European Union Directive 2010/63/EU). This project utilized 48 male C57BL/6J mice (Charles River Labs, Barcelona, Spain) housed at the University of Coimbra UC-Biotech Bioterium. They were accommodated in a well-ventilated room under a 12-h light/dark cycle. Before the beginning of the experiment, mice that were 8 weeks old were allowed to accommodate for 2–4 weeks with free access to standard chow and water. Four mice were kept in each cage.

The feeding trial was conducted for 18 weeks, following the 2 weeks of accommodation. Half of the mice were given a standard chow (Mucedola s.r.l. 223426), and the other half were provided with a personalized high-fat chow (Mucedola s.r.l. 223425) (see Appendix A for nutritional information). The chow cohorts were each subdivided into two groups, one where the drinking water was supplemented with 30% (*w*/*v*) of a 55/45 mixture of fructose and glucose corresponding to the nominal proportions of these sugars in the most widely used high-fructose corn syrup (HFCS) formulation, and the other with conventional drinking water. Consequently, there were 4 groups, each corresponding to one type of diet: standard chow control diet (SC), high fat (HF), standard chow plus the fructose corn syrup formulation (high sugar—HS) and high fat plus the fructose corn syrup formulation (high fat-high sugar—HFHS). Mice were randomly allotted to each group. At the beginning of the final evening of the feeding trial, all animals were administered 99.8% deuterated water (^2^H_2_O, CortecNet, Ulis, France) containing 0.9 mg/mL NaCl intraperitoneally to achieve a loading dose of ~4 g per 100 g of body weight. They also had their drinking water replaced with 5% ^2^H-enriched water. For the animals whose drinking water was supplemented with the HFCS formulation, the fructose component was enriched to 20% with [U-^13^C]fructose (Omicron Biochemicals, Inc., South Bend, IN, USA). Figure 2 schematically describes the design of the animal experiment. On the following morning, mice were deeply anaesthetized with ketamine/xylazine and sacrificed by cardiac puncture followed by cervical dislocation. Hepatic and arterial blood was collected and centrifuged to isolate plasma, livers were freeze-clamped, and the samples were stored at −80 °C until further processing.

### 2.2. Hepatic Triglyceride and Glycogen Extraction, Derivatization and Purification to Monoacetone Glucose (MAG)

Livers were finely ground under liquid nitrogen and lipids were extracted using the methyl tert-butyl ether (MTBE) lipid extraction protocol with triglycerides purified by solid-phase extraction (SPE) as previously described [35,38]. With this protocol, glycogen is present in the sedimented insoluble material. This precipitate was mixed with 30% KOH (2 mL/g of liver tissue) at 70 °C until complete dissolution, then treated with 6% Na_2_SO_4_ (1 mL/g of liver weight).

Glycogen was precipitated with ethanol (7 mL/g of liver weight). Samples were centrifuged for 10 min at 955× *g* and the pellet was resuspended in 5 mL of distilled water. An initial aliquot of 100 μL was collected for liver-free glucose quantification. The remaining samples were incubated with 120 units of amyloglucosidase from *Aspergillus niger* (Glucose-free preparation, Fluka biochemika, St. Gallen, Germany) for 6 h at 55 °C. They were centrifuged for 10 min at 1301× *g* and their supernatants collected. A final 100 μL aliquot was collected for glucose quantification following glycogen hydrolysis to determine the concentration of glycogen-derived glucose units. Samples were lyophilized, then resuspended with vigorous stirring in 5 mL of 2% ^2^H-enriched acetone containing 2% enriched ^2^H_2_SO_4_ (4% *v/v*) overnight at room temperature. The reaction was quenched by adding 5 mL of water. Samples were then incubated at 40 °C for 5 h at pH 7–8.

Lastly, they were lyophilized, and glucose was extracted using 5 mL boiling ethyl acetate. The supernatant was collected and evaporated. A set of 500 mg Discovery^®^ DSC-18 SPE columns (Sigma-Aldrich Co., Burlington, MA, USA) was washed with 3 mL of acetonitrile and 10 mL of water. Each sample was loaded in 1.5 mL of water in the column and the MAG fractions were eluted using ~2.5 mL of 10% acetonitrile/90% water (*v/v*). The MAG fraction was left to dry in the fume hood overnight and lyophilized until further analysis.

Hepatic glycogen was quantified from the initial and final aliquots from the glycogen hydrolysis protocol described above. These were assayed for glucose with a Miura 200 spectrophotometric analyzer (I.S.E. S.r.l.; Guidonia, Italy). Quantifications of glucose were performed in a fully-automated analyser using its dedicated reagent kit (A-R0100000601) and compatible vials according to the protocol described by Keppler and Decker [39]. To assess the number of glucosyl units in the glycogen molecule, the glucose measured in the initial aliquot was subtracted from that measured in the final aliquot. The glycogen concentration was expressed as μmol of glucosyl units per gram of wet tissue weight. Hepatic triglyceride was quantified from the extracted lipid by ^1^H-NMR [16].

### 2.3. Liver Histology

Liver samples were fixed in 10% buffered neutral formalin and processed for light microscopy using a tissue processor (Automated Leica Tissue Processor—TP1020). Tissue samples were dehydrated in a series of ethanol solutions (70%, 96% and 100%, each twice for 90 min), xylene (Klinipath, twice for 90 min) and then embedded in paraffin at 70 °C (Diapth, twice for 90 min). Next, the formalin-fixed paraffin-embedded tissue blocks were sectioned (Microm Microtome HM200) to the desired thickness of 3 μm and affixed onto glass slides for further staining. Before each staining protocol, slides were deparaffinized in xylene for 15 min and hydrated in series of ethanol solutions (100%, 96%, and 70%, each for 5 min) and distilled water (5 min). For routine histology haematoxylin and eosin staining, slides were stained with Harris Haematoxylin (Merck) for 10 min, dipped in 1% acid alcohol (3 dips of 1 s each), washed in running tap water for 5 min, dipped in ethanol 70% and then in alcoholic Eosin Y (Sigma-Aldrich) for 4 times of 1 s each. Finally, slides were dehydrated in a series of ethanol solutions (70%, 96% and 100%, 30 s each), cleared in xylene and allowed to mellow before mounting with a resinous mounting medium. For Masson’s Trichrome staining, slides were re-fixed in Bouin’s solution (Sigma-Aldrich) at 56 °C for 1 h, rinsed in running tap water for 5 min, stained in Weigert Iron Haematoxylin working solution (Merck) for 10 min, and again rinsed in running tap water for 10 min. Then, slides were stained in Biebrich scarlet-acid fuchsin solution (Atom Scientific, Merck, Rahway, NJ, USA) for 10 min and differentiated in Phosphomolybdic acid (Bio optica) for 15 min. Slides were directly transferred to 39 aniline blue solution for 10 min, washed in distilled water, dehydrated very quickly in ethanol (96%, 100%), cleared in xylene and mounted with a resinous medium.

Liver steatosis and fibrosis were assessed by a pathologist according to published guidelines and given a NAFLD Activity Score (NAS) by considering the degree of steatosis (0–3) and steatosis location, ballooning (0–2), inflammation (0–3), fibrosis (0–3), and the presence/absence of microvesicular steatosis, microgranulomas, large lipogranulomas, portal inflammation, and megamitochondria [40].

### 2.4. NMR Spectroscopy

Proton-decoupled ^2^H-NMR spectra of MAG were acquired at 50 °C as previously described using a Bruker Avance III HD 500 MHz spectrometer (Bruker Co., Bremen, Germany) with a ^2^H-selective 5 mm probe incorporating a ^19^F-lock channel [41]. Proton-decoupled ^2^H-NMR spectra of triglyceride were acquired at 25 °C with the same NMR system as previously described [35]. To determine deuterium enrichment of the body water, 10 μL plasma samples were analyzed by ^2^H-NMR as previously described [35,36]. MAG and triglycerides obtained from mice that received both [U-^13^C]fructose and ^2^H_2_O were also analyzed by ^13^C-NMR following ^2^H-NMR spectroscopy. MAG samples were evaporated and resuspended in 0.2 mL 99.8% ^2^H_2_O and triglycerides were evaporated and resuspended in 0.5–1.0 mL 99.9% CDCl_3_. For both analytes, 0.2 mL aliquots were pipetted into a 3 mm diameter NMR tube. Proton-decoupled ^13^C-NMR spectra were obtained at 25 °C with a Varian VNMRS 600 MHz spectrometer (Agilent Technologies, Santa Clara, CA, USA) using a 3 mm broadband probe with z-gradient. A pulse angle of 60° and acquisition time of 4.0 s followed by 0.1 s of pulse delay was used. The number of FIDs collected per sample ranged from 1640 to 13,128. The summed FID was processed with 0.2 Hz line-broadening and zero-filled to 131,072 points before Fourier transform [35,36].

### 2.5. Metabolic Flux Calculations

Direct and Indirect pathway contributions to hepatic glycogen synthesis for all four groups of mice was estimated from the positional ^2^H-enrichments of glycogen as previously described [36]. The fractional synthetic rate of triglyceride fatty acid and glycerol as well as fractional rates of fatty acid elongation and desaturation were estimated for the 4 groups of mice from the positional ^2^H-enrichments of liver triglyceride as previously described [16,36]. Whole body adiposity was assessed from the ^2^H-enrichment of body water as previously described [42]. For the mice that received [U-^13^C]fructose, an integrated analysis of glycogen and triglyceride ^2^H and ^13^C-enrichments was additionally applied to provide estimates of some of the major fluxes connecting exogenous sugar metabolism and DNL [37,43].

### 2.6. Statistical Analysis

Data organization, flux calculations and some preliminary corrections were performed in Microsoft Excel (Office 365, 2002) and the statistical analysis was conducted in GraphPad Prism 9.0.0. (GraphPad, Inc., San Diego, CA, USA). All datasets were evaluated for their normality and homoscedasticity. In experiments with only two condition groups that passed both normality and homoscedasticity tests, unpaired *t*-tests were used. In case the distribution was normal but there was lack of homoscedasticity, a Welch’s correction was applied. When normality was not verified, a Mann-Whitney test was used. In experiments with more than 2 conditions, one-way ANOVA tests were opted for. In that case, if the distribution was normal and homoscedasticity was verified, then an ordinary one-way ANOVA with Tukey’s post-test was used. If normality but not homoscedasticity was found, then a Welch’s ANOVA with Dunnett’s T3 multiple comparison post-test was done. In cases where data distributions were not normal, a Kruskal-Wallis with Dunn’s multiple comparison post-test was performed. The difference between two groups was considered statistically significant when the *p*-value was below 0.05. The decision to consider a point as an outlier was based on the ROUT test (Q = 1%). All data shown in Tables are represented as means accompanied by the standard deviation. All data depicted in graphs are shown as box and whiskers plots with bars representing minimum and maximum values.

## 3. Results

### 3.1. Effects of the Diets on Body Weight, Adiposity and Liver Triglyceride (TG)

Following an 18-week feeding trial, diet composition affected both weight gain and adiposity. Fat-rich diets (HF and HFHS) led to increased weight gain, whereas an excess of sugar intake alone (HS) caused no statistically significant change compared to control (SC) (Figure 3A). Significant differences in body water ^2^H-enrichment data between the diet groups (Appendix A) informs differences in whole body adiposity that showed a similar pattern to that of weight gain (Figure 3B). Fat intake (HF and HFHS) significantly increased the degree of adiposity compared to SC (Figure 3B). Thus, the additional weight gains associated with HF and HFHS diets can be attributed to increased whole body lipid stores.

Total hepatic TG as well as TG-non-essential fatty acids were modified by both fat and sugar intake. While total TG levels were significantly higher for the fat and/or sugar feeding conditions compared to SC (Figure 3C), HFHS mice had the highest concentration of TG-non-essential fatty acids per gram of liver (Table 1). HF and HFHS did not have significantly different liver triglyceride concentrations from each other, but all were significantly higher compared to SC (Figure 3C). These results suggest that elevated dietary fat and sugar have an additive effect on hepatic TG concentration. Liver histological analyses (Figure 4) were consistent with the above data and showed different degrees of steatosis with little or no accompanying fibrosis or inflammation. SC mice had hepatocytes with clear cytoplasm and centrally located nuclei compatible with glycogen accumulation. HF mice showed moderate steatosis mostly in zone 3 together with mild ballooning. There were no indications of fibrosis or inflammation. HS mice presented mild macrovesicular steatosis and mild ballooning mostly in zone 1 with no fibrosis or inflammation. HFHS mice presented severe steatosis and moderate ballooning as well as very mild perisinusoidal fibrosis. Average NAS scores were 0.0 ± 0.0 (SC), 3.3 ± 1.03 (HF), 2.0 ± 1.10 (HS) and 5.2 ± 1.92 (HFHS). 

In addition to alterations in liver TG levels, there were also significant divergences in hepatic TG fatty acid profiles, especially for essential and monounsaturated fatty acids (Appendix A). Compared to SC, HS mice had significantly diminished proportions of ω-3 fatty acids with a strong tendency for a reduction in linoleic acid, likely reflecting a dilution of calories obtained from the chow by the supplemented sugar. In contrast, HF mice had a significantly higher fraction of linoleic acid compared to both SC and HS mice, with comparable amounts of ω-3 fatty acids to SC mice. HFHS mice had similar proportions of essential fatty acids to SC mice. For monounsaturated fatty acids, HS mice had a significantly higher abundance of oleic acid compared to the other three groups, while palmitoleic acid was significantly more abundant in SC and HS mice compared to HF and HFHS. With the exception of palmitoleic and oleic acid, the fatty acid profile of HFHS mice was generally intermediate to those of HS and HF.

For all diet groups, there was a markedly non-uniform distribution of fatty acid classes between the *sn*2 and *sn*1,3 glyceryl sites with the majority of saturated fatty acids bound to the *sn*1,3 positions with only minor quantities at the *sn*2 site. Conversely, both mono- and polyunsaturated fatty acids were more abundant in the *sn*2 compared to *sn*1,3 sites (Appendix A). These positional distributions resemble those observed in human adipose tissue triglycerides [44].

### 3.2. Effects of the Diets on Hepatic Triglyceride Synthesis

Liver triglyceride was enriched with deuterium (^2^H) in all four mouse groups with HS and HFHS mice additionally enriched with ^13^C-isotopomers derived from metabolism of [U-^13^C]fructose (Appendix A). This allowed overall rates of glycerol and fatty acid synthesis to be estimated as well as the contribution of the fructose component of the fructose/glucose supplement to these synthetic fluxes for HS and HFHS mice. Because the diets resulted in significant divergences in hepatic TG concentrations, the lipogenic activities estimated from the ^2^H-enrichment levels were expressed both as fractional synthetic rates (FSR), and as newly synthesized lipid equivalents per gram of liver (Table 1).

Both SC and HS mice had fractional DNL rates that were similar to those measured using the same experimental design and analytical methodology [16,35]. Mice fed high-fat chow (HF and HFHS) had significantly lower fractional DNL rates compared to those fed standard chow (SC and HS). Nevertheless, the supplementation of high-fat chow with high-fructose corn syrup formulation was associated with a significant and almost 2-fold increase in fractional DNL rates.

Fatty acid elongation rates (EL) were significantly lower for both HS and HFHS compared to SC mice and tended to be lower in HF compared to SC mice (*p* = 0.09) with no significant differences between HS and either HF or HFHS mice (Table 1). Desaturation rates (DS) were significantly lower in HF compared to SC mice. While supplementation of either standard chow or high-fat diets with HFCS-55 did not result in significantly different desaturation rates, there were tendencies towards opposite effects for each diet, i.e., a decrease for HS over SC and an increase for HFHS over HF. Consequently, there were no significant differences in desaturation rates between HS and HFHS mice (Table 1).

The FSR of triglyceride glycerol exceeded those of DNL for all diet conditions and represented a substantial portion (~30–50%) of the entire liver TG pool (Table 1). The highest rates were found for SC mice, with HF mice showing a tendency for lower values compared to SC (*p* = 0.08) while both HS and HFHS had significantly lower rates compared to SC. When the parameters of lipid synthesis were expressed in terms of fatty acid equivalents per gram of liver, DNL values for SC and HF were similar, while those of HS and HFHS were significantly higher (Table 1). Similar trends were also observed for fatty acid elongation and desaturation rates.

^13^C-isotopomer analysis was applied to the TG samples to determine the contribution of the dietary fructose component to newly-synthesized triglyceride fatty acids and to glycerol [25] in HS and HFHS mice (Table 2). As shown in the example of Figure 5, triglyceride ^13^C-NMR spectra had natural-abundance ^13^C-singlet signals from mice that did not receive ^13^C-tracer, as shown by the example from an SC mouse. For mice that were provided with [U-^13^C]fructose, the spectra also contained doublet signals arising from ^13^C-^13^C-coupling and representing enrichment from the [U-^13^C]fructose precursor. These allowed a robust quantification of saturated fatty acid and oleic acid enrichments but the palmitoleic acid doublet signals, while discernible in most spectra, could not be reliably quantified. In both HS and HFHS spectra, the palmitoleic acid doublet/singlet ratio had a similar aspect to that of oleic acid. For both HS and HFHS mice, fructose contributed a significantly higher fraction of acetyl-CoA to the synthesis of saturated fatty acids compared to oleic acid. Exogenous fructose also contributed a substantial fraction of newly synthesized glycerol-3-phosphate, with a strong tendency toward a significantly higher contribution in HS compared to HFHS mice.

### 3.3. Effects of the Diets on Hepatic Glycogen Synthesis

Hepatic glycogen concentrations did not vary significantly between the different diets and ranged from 70–90 mg/g of wet weight of liver tissue (Appendix A). Glycogen was enriched with ^2^H in all four mouse groups with HS and HFHS mice additionally enriched with ^13^C-isotopomers derived from metabolism of [U-^13^C]fructose (Appendix A). Glycogen turnover was essentially complete for all four groups as seen by fractions of newly synthesized glycogen of 106 ± 18, 116 ± 13, 97 ± 20 and 96 ± 9% for SC, HF, HS and HFHS, respectively. The sources of hepatic glycogen synthesis after overnight feeding are described in Table 3 and were significantly influenced by diet. For SC mice, the Direct pathway was the dominant source accounting for over two-thirds of newly synthesized glycogen. For HF mice, the Direct pathway contribution was significantly lower compared to SC, accounting for about half of the glycogen. Both HS and HFHS mice had Direct pathway contributions that were intermediate to SC and HF. The gluconeogenic Indirect pathway supplies the balance of hepatic glycogen synthesis. Its contributions, resolved as originating via anaplerotic Krebs cycle metabolism of substrates such as pyruvate (KC), or from substrates such as fructose and glycerol entering at the level of triose-P (TP), were also influenced by the different diets. Compared to SC mice, HF mice had a significantly higher Indirect pathway contribution via KC, while for HS the dominant Indirect pathway sources were via TP. HFHS mice had equivalent Indirect pathway contributions from both KC and TP sources. For HS and HFHS mice, the contribution of the fructose component of high-fructose corn syrup to glycogen synthesis was found to be 17 ± 9 and 11 ± 6%, respectively, with almost all the carbons being metabolized via TP. For HS mice, fructose accounted for most TP precursors (17/24, or 71%) while for HFHS mice, it contributed just over half (11/20 or 55%).

### 3.4. Integrated Analysis of Substrate Fluxes into De Novo Lipogenesis

The integrated analysis combines the measurements of hepatic glycogen and triglyceride ^2^H-enrichments and ^13^C-isotopomers from ^2^H_2_O and a single [U-^13^C]sugar (in this case [U-^13^C]fructose) to provide an overview of the fluxes connecting glucose-6-P and DNL. It incorporates measurements of pentose phosphate (PP) activity as well as contributions of glycolytic and non-glycolytic substrates to the lipogenic acetyl-CoA pool (Figure 6). Overall, HS and HFHS mice showed similar substrate selection profiles to those reported in our previous studies [37,43]. Most notably, despite the high abundance of dietary sugar, about half of all acetyl-CoA recruited for DNL originated from substrates that were not metabolized via glycolysis such as short chain fatty acids and ketogenic amino acids. The main differences between HS and HFHS was in the sources of triose-P for triglyceride glycerol synthesis, with the fractional contribution of fructose being significantly lower in HFHS compared to HS. Also, the contribution of dietary glucose to triose-P was significantly higher in HFHS compared to HS mice but gluconeogenic precursors nevertheless accounted for most triose-P sources for both diets. PP fluxes relative to acetyl-CoA lipogenesis were not different between the two groups. 

To summarize, fat intake was the main contributor to increased lipid stores leading to weight gain, even though sugar intake aggravated the degree of steatosis, especially in HFHS. The high fat diet reduced DNL rates, while sugar intake caused a significant increase in DNL, especially in HFHS. The rate of de novo glycerol synthesis exceeded DNL for all diets with exogenous fructose substantially contributing to the newly synthesized glycerol-3-phosphate pool. Elongation was lowered in high fat diets and fructose significantly increased the fraction of acetyl-CoA used as precursor for saturated fatty acid synthesis. Despite high abundance of dietary sugar, roughly half of the acetyl-CoA recruited for DNL originated from short chain fatty acids and ketogenic amino acids. Hepatic glycogen concentrations did not vary significantly between the different diets and glycogen turnover was complete for all groups. The direct pathway was dominant in SC and the lowest for HF, even though it still accounted for half of the newly synthesized glycogen. In HF, the main source of intermediates to the indirect pathway was anaplerotic Krebs cycle, whereas for HS the dominant source of intermediates was TP precursors. All fructose entered the indirect pathway at the level of Triose-P, representing most of the TP in HS.

## 4. Discussion

### 4.1. NAFLD Profiles of Our Mice Models

The onset and development of MASLD involves the accumulation of lipid droplets inside hepatocytes, resulting in isolated steatosis. Subsequently, the liver may develop inflammation and fibrosis thereby progressing into Metabolic Dysfunction-Associated Steatohepatitis (MASH)—a more severe form of the disease, which in turn can lead to cirrhosis and/or hepatocellular carcinoma [2,45].

High fat diet, high fat with cholesterol and refined sugars (Western diet) and/or high fat with refined sugars in drinking water (American lifestyle diet) are very effective in causing MASLD with a mild histological phenotype [3]. This is in line with our models of high fat, high sugar in drinking water, or a combination of both, which recapitulated different degrees of steatosis without significantly inducing fibrosis or inflammation, thus resembling the early stage of MASLD.

To the extent that hepatic carbohydrate and lipid metabolism are modified by pro-inflammatory factors such as TNF-α, the substrate sources for hepatic glycogen and triglyceride synthesis and the activity of the hepatic PP pathway could be substantially different in a MASH compared to a MASLD setting [46,47]. In recognition that the wild-type mice strains in general use are relatively resistant to the development of MASH through overfeeding of dietary sugar and/or fat alone, supplementation with hepatotoxic agents such as chloroform or trans-fats have been used to induce a more MASH-like hepatic profile. Alternatively, the offspring of a stable isogenic cross between C57BL/6J and 129S1/SvImJ mice fed with high fat/high fructose diet were found to develop MASH with fibrosis over the same timescale as our feeding trial [48,49,50]. The use of choline-deficient diets has been reported to rapidly induce significant fibrosis, yet mice experience significant weight loss, therefore these models fall short at accurately represent the metabolic burden of MASLD [3].

### 4.2. Interactions of Dietary Sugar and Lipid in Promoting Steatosis

Oversupply of either lipid or sugar to the liver results in elevated levels of hepatocellular triglyceride. The molecular controls of sugar and lipid metabolism are strongly linked, and in many cases involve reciprocal regulation in the metabolism of one nutrient over the other. Most notably, high intracellular levels of fatty acid metabolites suppress glycolysis and the oxidation of carbohydrate to acetyl-CoA [51,52]. Conversely, an increase in DNL activity attenuates fatty acid oxidation via malonyl-CoA-mediated inhibition of mitochondrial long-chain fatty acid uptake [20]. The sugar supplement had an excess of fructose, whose metabolism to triose-P bypasses key lipid-mediated control mechanisms of carbohydrate metabolism at the levels of glucose-6-P and fructose 1,6-P_2_ [17,53,54]. In terms of liver TG concentrations and overall adiposity, high levels of dietary sugar and fat were additive, as shown in Figure 3. For HFHS mice, the high fat chow resulted in an increased accumulation of hepatic TG from dietary lipid as seen by the increased fraction of linoleic acid in HFHS compared to either SC or HS mice. Meanwhile, HFHS mice had significantly higher fractional DNL rates compared to HF animals that accounted for ~10% of hepatic TG. In addition to this direct contribution, it is possible that this increased fractional DNL activity also indirectly contributed to excessive lipid levels by inhibiting mitochondrial oxidation of long-chain fatty acids. The FSR data expressed in terms of liver fatty acid concentrations (Table 1) suggest that this indirect effect may indeed have had a far greater impact in potentiating steatosis. While the concentration of TG fatty acids attributable to DNL was ~30 μmol/g liver higher in HFHS compared to HF mice, the total concentration of non-essential fatty acids was ~300 μmol/g higher.

Contrarily to Duarte et al., 2014 [16], we demonstrated a tendency towards lower elongation rates accompanied by significantly lower desaturation rates in HF compared to SC (Table 1). These can be explained by the differences in the composition of the fats used in each chow. In the current study, the high fat diet is 30% fats, of which 59% polyunsaturated, 24% monounsaturated and only 17% saturated fatty acids. In opposition, in Duarte et al. the chow was 60% fats, of which the majority was lard (310 g/kg of chow) and soybean oil (30 g/kg of chow), which are mostly saturated fatty acids. In this way, the high fat diet used in our current study was less prone to promote either elongation or desaturation.

### 4.3. Effects of Diet on Sources of Hepatic Glycogen Synthesis 

Recent studies have shown that hepatic glycogen concentrations *per se* are linked with both hepatic metabolic status as well as central control of food intake [32,33,55]. Since we did not see any significant differences in hepatic glycogen concentrations between the different diet treatments, we cannot fully determine any effects of the diets on these parameters via modulation of glycogen levels. Nonetheless, the diets did have a significant influence on the sources of hepatic glycogen synthesis during a single nocturnal feeding period [56]. In HF mice, the dominance of Indirect over Direct pathway is consistent with lower amounts of dietary glucose precursors and/or impaired glucokinase activation secondary to hepatic insulin resistance. Supplementation of high-fat chow with HFCS-55 restored direct pathway contribution, possibly via glucokinase activation by fructose-1-P, generated by the initial phosphorylation of fructose via ketohexokinase [57]. In HS mice, fructose was also efficiently utilized as an Indirect pathway substrate as seen by its strong contribution to the TP precursors and an overall increase in TP over KC Indirect pathway fractions. This is consistent with our previous observations of mice fed diets with various levels of fructose [36,58]. Its contribution was blunted in HFHS mice, possibly in part because of competition from glycerol derived via lipolysis of dietary triglyceride.

### 4.4. Effects of Diets on the Sources of Hepatic Triglyceride Synthesis

The strong coordination of both hepatic glucose and fructose metabolism with DNL is mediated by transcription factors such as ChREBP and SREBP1c and is relatively unperturbed by insulin resistance [59,60,61,62]. Fructose is quickly cleared from the blood by the liver and phosphorylated by fructokinase, thereby avoiding the regulation of PFK-1 and insulin. Moreover, fructose has been shown to promote DNL, inhibit fatty acid β-oxidation and potentially inhibit insulin signaling [63]. Fructose metabolism yields triose phosphates that feed gluconeogenesis to be ultimately incorporated into glycogen, while also generating a significant amount of acetyl-CoA to enter the lipogenic pathway [62,64]. Our data indicate that even during high sugar feeding, about half of the lipogenic acetyl-CoA originated from non-glycolytic sources. Since the contribution of glycolytic and non-glycolytic substrates to lipogenic acetyl-CoA was estimated from the dilution of the ^13^C-tracer between hepatic triose-P and the acetyl-CoA moiety of newly synthesized fatty acids, these non-glycolytic substrates could have included pyruvate precursors such as alanine, glutamine, proline, lactate, ketogenic amino acids, and acetate generated from fermentation activity of colonic microorganisms [28,65,66]. Recent studies have established the importance of intestinal acetate as a lipogenic substrate and that depletion of microbiota or blockage of hepatic acetate metabolism via silencing of acetyl-CoA synthase 2 resulted in a significant attenuation of DNL [67,68]. Zhao et al. demonstrated that a portion of dietary fructose is converted to lipid via microbial acetate formation—in our study this would have been counted as part of the glycolytic contribution (see Discussion Section 4.6—Study limitations) [67]. Thus, to the extent that the dietary [U-^13^C]fructose was converted to lipid via microbial acetate, the contribution of non-glycolytic sources to DNL was underestimated.

Our study demonstrated that fructose contributes to TG synthesis both through DNL and TG-glycerol synthesis. Furthermore, our study confirms our previous observation that fructose contributed a larger fraction of acetyl-CoA to the synthesis of saturated compared to monounsaturated fatty acids [35]. While the proportion of dietary monounsaturated fatty acids was slightly higher than that of saturated fatty acids (24 versus 17% of total dietary lipid) for both standard chow and high-fat chow, we believe this does not fully explain the difference in the ^13^C-enrichments of saturated *versus* monounsaturated liver triglyceride fatty acids from [U-^13^C]fructose. We speculate that these observations might also, in part, reflect the hepatic zonation of fructose metabolism giving rise to different enrichments of acetyl-CoA pools between periportal, pericentral and perivenous regions. Studies with [1,2-^13^C_2_]acetate infused into livers of conscious dogs via transhepatic catheters demonstrated a marked heterogeneity of lipogenic acetyl-CoA enrichment attributable to hepatic zonation of acetate metabolism [69]. Given that fructose is efficiently extracted by the liver, this infers a concentration gradient that decreases between periportal and perivenous zones possibly resulting in different contributions to periportal *versus* pericentral and perivenous acetyl-CoA pools. To the extent that the enzymes of fatty acid synthesis and desaturation are also heterogeneously distributed in the hepatic lobule, this might provide an explanation for the different ^13^C-enrichments of saturated and monounsaturated fatty acids from dietary [U-^13^C]fructose. While acetyl-CoA carboxylase and fatty acyl synthase are considered to more prevalent in pericentral compared to periportal hepatocytes, fatty acid esterification by glycerol-3-P to form lysophosphatidic acid is more concentrated in periportal hepatocytes [70,71]. However, it is not known whether fatty acyl synthase and desaturase enzymes have the same or different zonal distributions.

### 4.5. Stoichiometry of PP and DNL Fluxes

One important consequence of lipogenic acetyl-CoA recruitment from non-glycolytic substrates is the decoupling of lipogenic acetyl-CoA formation from PP flux. In the conversion of glucose-6-P to palmitate, for each mol of glucose-6-P converted to acetyl-CoA, 0.29 mol need to undergo PP oxidation in order to generate the required NADPH equivalents [37]. If it is assumed that the PP is the sole source of lipogenic NADPH, then, in order to provide NADPH for DNL of non-glycolytic substrates such as acetate, the proportion of glucose-6-P oxidized by the PP would need to be further increased. In fact, given that half of the lipogenic acetyl-CoA was derived from non-glycolytic substrates (Figure 5), then for each mol of glucose-6-P recruited for DNL, 0.58 mol would need to be oxidized by the PP—corresponding to a fractional PP rate of 0.58/1.58 or 37%. Experimentally, the ratio between PP fluxes and lipogenic carbohydrate utilization was found to be much smaller as shown by the data in Figure 5: 5/(5 + 53) = 9% for HS and 6/(53 + 6) = 10% for HFHS. Within the scope of lipogenic glucose-6-P fluxes, this level of PP activity is well below that required to fulfil the NADPH demands of DNL. This could be mitigated by fluxes through other NADPH-producing pathways such as cytosolic NADP-malic enzyme and cytosolic NADP-isocitrate dehydrogenase. Also, if the glycolytic flux was higher than that of pyruvate dehydrogenase (i.e., resulting in net lactate production), then for a given fractional PP activity, more NADPH would be available per equivalent of lipogenic acetyl-CoA. Interestingly, human studies show that a significant fraction of fructose carbons is recovered in circulating lactate independently of whether or not it was accompanied by glucose [72]. It is not known what fraction of this fructose underwent PP oxidation prior to lactate formation.

### 4.6. Study Limitations

There are several important methodological limitations of our approach that need to be considered. Assuming that hepatic TG levels remain relatively constant throughout the overnight interval of tracer administration, for our measurements of substrate and metabolic pathway contributions to the hepatic triglyceride pool to be translated into absolute flux rates then lipid outflows (i.e., hepatic long-chain fatty acid oxidation and triglyceride clearance via VLDL efflux) need to be also measured. While the product of fractional synthetic rates and liver TG concentrations provides some additional insight into interactions between the different sources of liver TG—most notably the apparent leveraging of total lipid content by DNL in the setting of high fat and high sugar feeding—these quantities do not directly translate into absolute flux rates. Our estimates of fractional lipid fluxes had relatively large standard deviations. This may reflect the fact that the animals were allowed to feed freely during the overnight labelling period so the quantity and/or frequency of chow and sugar ingested was likely different between each animal. Given that hepatic lipid synthesis is highly controlled by insulin—whose secretion in turn is acutely tuned to food intake—any individual variability in food intake likely contributed to the variance in the lipid synthesis flux estimates. 

Microbial fermentation of [U-^13^C]fructose results in the formation of [U-^13^C]acetate [46], whose incorporation into DNL is indistinguishable from that of [U-^13^C]acetyl-CoA derived from hepatic [U-^13^C]fructose metabolism. Thus, the contribution of glycolytic substrates to lipogenic acetyl-CoA formation may be overstated with those of non-glycolytic sources correspondingly understated. In addition, conversion of [U-^13^C]fructose to glucose by intestinal enterocytes and hepatic metabolism of this glucose to triose-P would generate identical ^13^C-triose-P isotopomers to those derived from hepatic fructolysis. Among other things, this would result in underestimates of Direct pathway contributions to glycogen synthesis and corresponding overestimates of Indirect pathway contributions from fructose. Estimates of PP flux are based on the sugar phosphates that are recycled back to fructose-6-P and glucose-6-P and do not take into account those pentose-P equivalents that were recruited for nucleotide biosynthesis. Thus, the PP estimates represent a lower limit of the real oxidative glucose-6-P flux.

Finally, this study only featured male C57BL/6J mice, which overlooks possible sex differences regarding sources of hepatic glycogen and triglyceride synthesis in response to different types of diet.

### 4.7. Conclusions

In preclinical mouse models, MASLD can be induced by excessive intakes of either fat or high-fructose corn syrup-like formulations, or a combination of both. When both are ingested in excess, there is an additive impact on liver TG levels. This is despite the inhibitory effect of hepatic fatty acid levels on the conversion of sugar to lipid via DNL. By quantifying the sources of hepatic glycogen and triglyceride synthesis, including the contribution of dietary fructose to these products, our study provides new insights on the interaction of dietary fat and sugar in promoting MASLD. Also, our study revealed that while postprandial hepatic glycogen repletion was not affected by the different lipogenic diets, the sources of glycogen synthesis were highly influenced by dietary fat and sugar composition with dietary fructose being a key precursor. Finally, our data show that a combination of high fat and sugar feeding was the most potent effector of hepatic steatosis. In this setting, DNL *per se* did not contribute a significant fraction of liver TG, but was associated with a substantial increase in total TG levels, possibly mediated in part by the inhibition of long-chain fatty acid oxidation mediated by malonyl-CoA.

## Figures and Tables

**Figure 1 nutrients-16-02186-f001:**
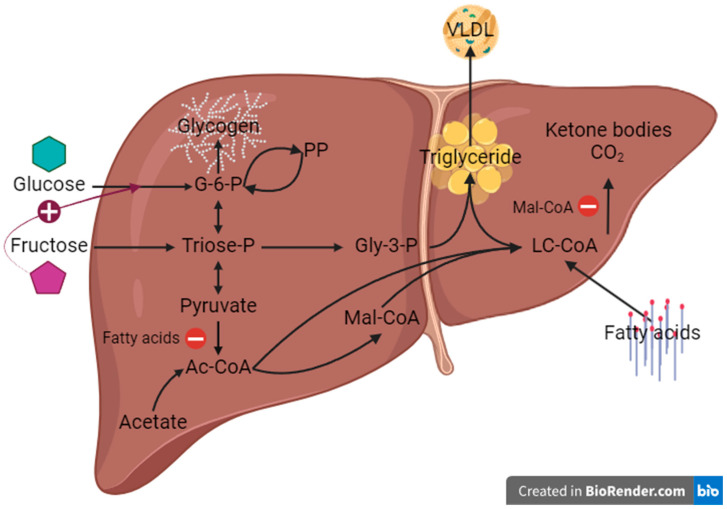
Principal hepatic lipid and carbohydrate fluxes under postprandial conditions in relation to the availability of fatty acids, glucose and fructose. Selected key influences of one substrate on the metabolism of the other are highlighted with colors: in red the inhibition of pyruvate dehydrogenase by fatty acids, in purple the inhibition of lipid oxidation by malonyl-CoA, and in blue the activation of glucokinase by fructose. For simplicity, only the main metabolic intermediates are shown. Ac-CoA = acetyl-coenzyme A, G-6-P = glucose-6-phosphate, Gly-3-P = glycerol-3-phosphate, LC-CoA = long-chain fatty acid-coenzyme A, Mal-CoA = malonyl-coenzyme A, PP = pentose phosphates, VLDL = very low-density lipoprotein (Created in BioRender.com).

**Figure 2 nutrients-16-02186-f002:**
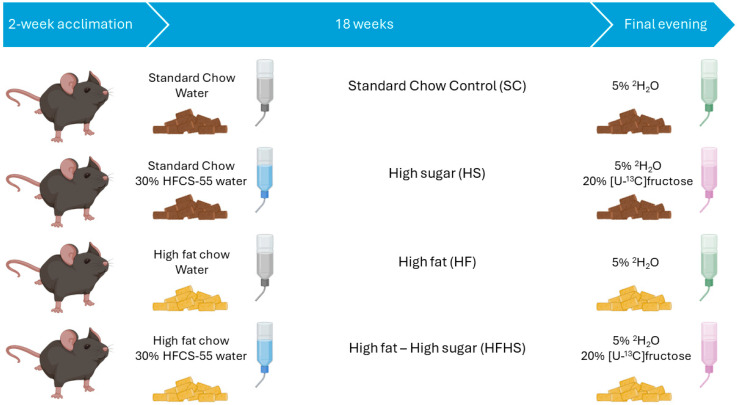
Schematic representation of the study design involving C57BL/6J mice (Charles River Labs, Barcelona, Spain). Dark pellets indicate standard chow and yellow pellets indicate high fat chow. Gray water bottles represent conventional water and blue water bottles indicate water supplemented with 30% (*w*/*v*) of a 55/45 mixture of fructose and glucose. Green bottles indicate 5% deuterium enrichment, whereas pink bottles indicate both 5% deuterium enrichment of the water component plus 20% [U-^13^C]fructose enrichment of the fructose component of the 30% (*w*/*v*) mixture of 55/45 fructose and glucose.

**Figure 3 nutrients-16-02186-f003:**
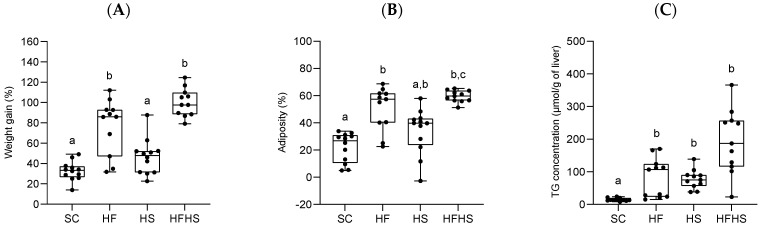
(**A**–**C**): Effects of 18 weeks of feeding with standard chow diet (SC, *n* = 12), high-fat chow diet (HF, *n* = 11), standard chow diet supplemented with HFCS-55 formulation in the drinking water (HS, *n* = 12), and high-fat chow diet supplemented with HFCS-55 formulation in the drinking water (HFHS, *n* = 11) on the body weight evolution (**A**), body weight fat fraction (**B**) and liver triglyceride concentration (**C**). One μmol/g of triglyceride is equivalent to approximately 0.97 mg/g. Data are shown as means accompanied by standard deviations. For each parameter, significant differences in values (*p* < 0.05) between the four diets are denoted by different letters.

**Figure 4 nutrients-16-02186-f004:**
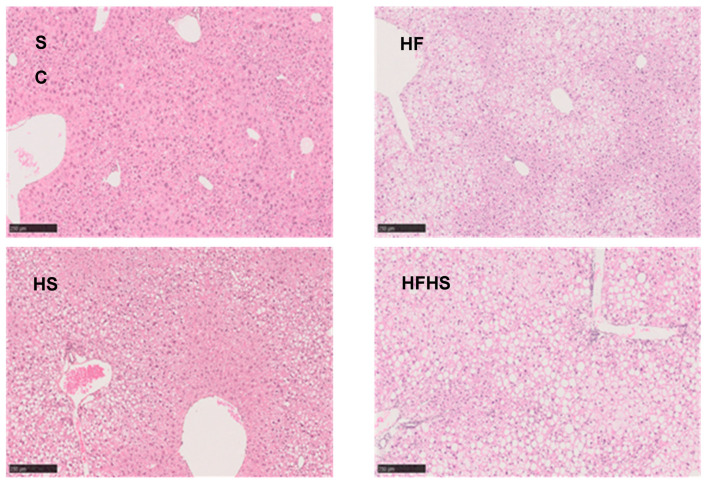
Liver histology—H&E staining of liver sections. Effects of 18 weeks of feeding with standard chow diet (SC, *n* = 12), high-fat chow diet (HF, *n* = 11), standard chow diet supplemented with HFCS-55 formulation in the drinking water (HS, *n* = 12), and high-fat chow diet supplemented with HFCS-55 formulation in the drinking water (HFHS, *n* = 11) on liver histology. The black bars are scales representing 250 μm.

**Figure 5 nutrients-16-02186-f005:**
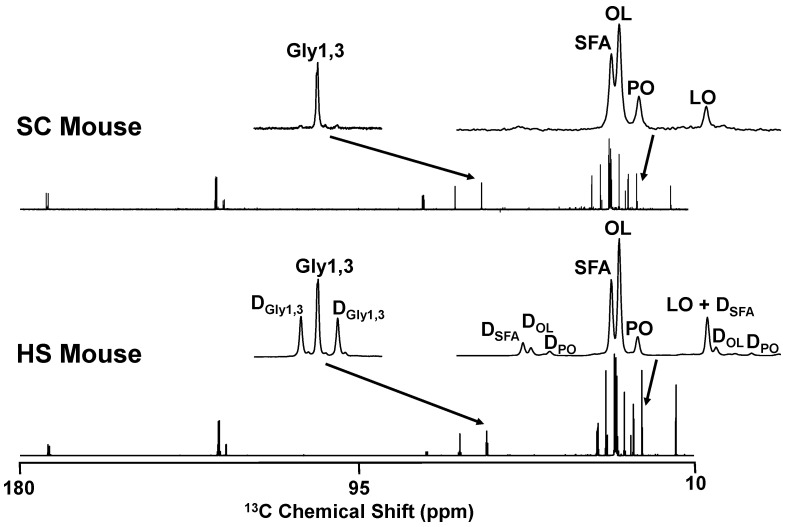
^13^C NMR spectra of liver triglyceride obtained from mouse fed standard chow (SC) and a mouse fed standard chow plus HFCS-55 formulation in the drinking water with the fructose component enriched to 20% with [U-^13^C]fructose (HS). The cluster of signals centered at 22.8 ppm representing the ω-2 signals of saturated fatty acids (SFA), oleic acid (OL), palmitoleic acid (PO) and linoleic acid (LO) and the signal from glycerol carbons 1 and 3 (Gly1,3) at 62.1 ppm are highlighted. In the HS spectrum, ^13^C-^13^C-spin coupled doublet signals from SFA (D_SFA_), OL (D_OL_), PO (D_PO_) and Gly1,3 (D_Gly1,3_) are also indicated.

**Figure 6 nutrients-16-02186-f006:**
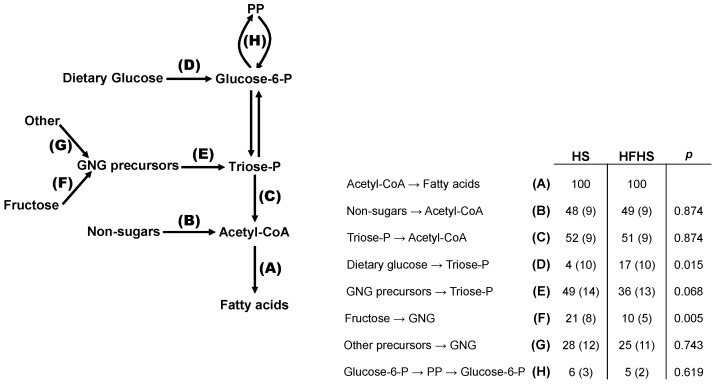
Integrated analysis of fractional lipogenic substrate and pentose phosphate (PP) pathway fluxes based on glucose-6-P isotopomer distributions (analyzed via glycogen) and triose-P isotopomer distributions (analyzed via triglyceride glycerol) [27]. Values are indexed to a DNL flux of 100 for HS and HFHS mice. Data are shown as means accompanied by standard deviations in parentheses. Due to exchange between triose-P and glucose-6-P, GNG precursor carbons can feed the glucose-6-P pool or can directly enter the triose-P pool (dashed line). The letters represent the following fluxes: A = conversion of acetyl-CoA to fatty acids via de novo lipogenesis; B = conversion of non-sugar substrates to lipogenic acetyl CoA; C = conversion of triose phosphates to acetyl-CoA; D = conversion of dietary glucose to triose phosphates via glucose-6-phosphate; E = conversion of gluconeogenic precursors to triose phosphate; F = conversion of fructose to gluconeogenic precusors; G = conversion of substrates other than fructose to gluconeogenic precursors; H = flux of glucose-6-phosphate through the oxidative and non-oxidative phases of the pentose phosphate pathway.

**Table 1 nutrients-16-02186-t001:** Liver triglyceride synthesis activity for the different diets resolved into rates of triglyceride glycerol synthesis (GLY) and rates of fatty acids synthesis (DNL) with elongation (EL) and desaturation (DS) rates. Lipid synthesis activity is expressed on the left-hand side as fractional synthetic rates (FSR) with GLY as a percentage of all triglyceride, DNL and EL as a percent of all non-essential fatty acids, and DS as a percentage of all monounsaturated fatty acids. On the right-hand side, lipid synthesis activity is expressed as the product of FSR and total liver triglyceride concentrations for GLY; FSR and concentrations of non-essential fatty acids (FA*_NE_*) for DNL and EL, and FSR and concentrations of monounsaturated fatty acids (FA*_MU_*) for DS. Data are shown as means accompanied by standard deviations in parentheses. For each parameter column, significant differences in values between the four diets are denoted by different superscript letters.

Diet	FSR (%)				FSR × [Lipid Species] (μmol/g Liver)						
	GLY	DNL	EL	DS	[TG]_total_	GLY	[FA*_NE_*]	DNL	EL	[FA*_MU_*]	DS
SC	50 (14) ^a^	17 (7) ^a^	16 (2) ^a^	16 (6) ^a^	18 (10) ^a^	9 (6) ^a^	43 (25) ^a^	8 (7) ^a^	8 (4) ^a^	32 (19) ^a^	6 (5) ^a^
HF	35 (18) ^a,b^	5 (2) ^b^	8 (4) ^a,b^	3 (1) ^b^	82 (60) ^b^	21 (11) ^a,b^	162 (121) ^b^	6 (4) ^a^	9 (5) ^a^	99 (75) ^a,b^	3 (2) ^a^
HS	30 (7) ^b^	15 (6) ^a^	10 (3) ^b^	9 (4) ^a,c^	95 (66) ^b^	28 (17) ^b^	275 (207) ^b^	38 (26) ^b^	26 (15) ^b^	209 (152) ^b,c^	18 (12) ^b^
HFHS	26 (8) ^b^	9 (3) ^c^	6 (3) ^b^	6 (3) ^b,c^	193 (98) ^b^	45 (19) ^b^	467 (245) ^c^	38 (18) ^b^	27 (16) ^b^	311 (170) ^c^	17 (8) ^b^

**Table 2 nutrients-16-02186-t002:** Percentage contributions of the fructose component of the high-fructose corn syrup formulation added to the drinking water of mice fed standard chow (HS) and mice fed high-fat chow (HFHS) to the acetyl-CoA pool of newly-synthesized saturated fatty acids (SFA), oleic acid (OL) and triglyceride glyceryl component (GLY). Values are expressed as means accompanied by their standard deviations in parentheses. For each parameter column, significant differences in values between the four diets are denoted by different superscript letters. Within each row, significant differences are denoted by asterisks.

Diet	Triglyceride Component		
	SFA	OL	GLY
HS	27 (15)	16 (10) * ^a^	40 (13) ^b^
HFHS	18 (9)	7 (5) ***	22 (11)

* *p* < 0.05 compared to SFA, *** *p* < 0.001 compared to SFA, **^a^**
*p* = 0.0036 compared to HFHS, **^b^**
*p* = 0.06 compared to HFHS (All Mann-Whitney tests).

**Table 3 nutrients-16-02186-t003:** Sources of hepatic glycogen synthesis in mice fed with standard chow (SC), high-fat chow (HF), standard chow plus high-fructose corn syrup formulation in the drinking water (HS) and high-fat chow plus high-fructose corn syrup formulation in the drinking water (HFHS). The sources are resolved into Direct and Indirect pathways, with the Indirect pathway further resolved into substrates metabolized via the anaplerotic pathways of the Krebs cycle (KC) or via triose-P intermediates (TP). For HS and HFHS mice, the fractional contribution of the HFCS-fructose component to glycogen synthesis via all indirect pathways (Fru*_total_*) is resolved into contributions via KC (Fru*_KC_*) and TP (Fru*_TP_*). Values are expressed as means accompanied by their standard deviations in parentheses. For Direct and Indirect pathway fractions column, significant differences in values between the four diets are denoted by different superscript letters.

Diet	Direct Pathway	Indirect Pathways					
		Total	KC	TP	Fru*_KC_*	Fru*_TP_*	Fru*_total_*
SC	71 (3) ^a^	29 (3) ^a^	19 (3) ^a^	10 (1) ^a^	N.D.	N.D.	N.D.
HF	52 (4) ^b^	47 (2) ^b^	34 (3) ^b^	12 (3) ^a,c^	N.D.	N.D.	N.D.
HS	64 (6) ^c^	36 (6) ^c^	12 (3) ^c^	24 (5) ^b^	1 (1)	16 (8) **	17 (9) **
HFHS	60 (3) ^c^	40 (3) ^c^	21 (4) ^a^	20 (3) ^b,c^	1 (1)	11 (5)	11 (6)

** significantly different from HFHS, *p* < 0.01 (Mann-Whitney test).

## Data Availability

The raw data supporting the conclusions of this article will be made available by the authors on request.

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
