# Peer review of "The Effects of Long-Term High Fat and/or High Sugar Feeding on Sources of Postprandial Hepatic Glycogen and Triglyceride Synthesis in Mice"

_nutrients, 2024, doi:10.3390/nu16142186_

Round 1
Reviewer 1 Report
Comments and Suggestions for Authors
This study systematically illustrated that the combination of long-term high fat and sugar feeding was the most potent effector of hepatic steatosis. Fructose was found to facilitate lipogenesis of saturated over unsaturated fatty acids and participate in the maintenance of glycogen levels. These findings provided a new insight of the relationship between DNL, fatty acid oxidation and hepatic triglyceride. Here are some problems need to be improved.
1. In the methods section, although the grouping of animals was completely described, a schematic diagram will be more helpful for better understanding the study design.
2. The reason for using 2H2O and 2H-enriched water instead of H2O should be mentioned.
3. At present, induced by high-fat diet is the most ideal method to build mice NAFLD model. This study shows that a combination of high fat and sugar feeding was the most potent effector of hepatic steatosis. So, whether the methods used in this study can provide a new approach to obtain NAFLD mice with better phenotypes.
4. In table 1, the standard deviations of FSR were relatively large, I wonder the possible reasons and whether it affects the calculation of the significance level between different groups.
5. In figure 2, A, B, and C were suggested to be placed at the top left corner.
Reviewer 2 Report
Comments and Suggestions for Authors
In this research paper, the Authors report the effect high fat (HF) and/or high sugar (HS) diet on sources of hepatic glycogen and triglyceride synthesis in male mice. To investigate this, the authors measured de novo lipogenesis (DNL) and glycogen levels in mice fed for 18 weeks with standard chow (SC), SC supplemented HS, HF, and HF with the HS supplementation (HFHS). Furthermore, for HS and HFHS mice, pentose phosphate (PP) fluxes and fructose contributions to DNL and glycogen were also measured. Overall, the presented data is interesting, well analyzed, and gives valuable data. The manuscript in general is well organized and written in good English.
I recommend revision with corrections as follows:
General comments:
Although the study was well designed and conducted, the main shortage of the study is the use of only male mice instead of conducting research on both sexes. The differences on sources of hepatic glycogen and triglyceride synthesis in response to different types of diet obtained in this study may not be the same in females and should be addressed as a limitation of the study.
Specific comments:
Abstract:
Lines 29-30: It is enough to just write “…, HF with HS supplementation (HFHS)”. You explained what HS means in the line before.
Introduction:
Line 69: remove “not least” from the sentence.
Line 75 -76: please rewrite: “in part because of the suppression by fatty acids of glucose oxidation to acetyl-CoA” to “partly due to fatty acid suppression of glucose oxidation to acetyl-CoA”
Line 117: “during unrestrained overnight feeding” correct to: “overnight ad libitum feeding “
Results:
Figure 3. The scale bar on the picture is not clear. If you can´t make it clearer, than write it in the figure legend.
I suggest moving the Table 1 to section 3.2. Effects of the diets on hepatic triglyceride synthesis, since most of results from this table is presented in this section.
